# Variations in Cold Resistance and Contents of Bioactive Compounds among *Dendrobium officinale* Kimura et Migo Strains

**DOI:** 10.3390/foods13101467

**Published:** 2024-05-09

**Authors:** Hexigeduleng Bao, Hainan Bao, Yu Wang, Feijuan Wang, Qiong Jiang, Hua Li, Yanfei Ding, Cheng Zhu

**Affiliations:** 1Key Laboratory of Specialty Agri-Product Quality and Hazard Controlling Technology of Zhejiang Province, College of Life Science, China Jiliang University, Hangzhou 310018, China; 13848880640@163.com (H.B.); wangyu90203@163.com (Y.W.); wfj0311@cjlu.edu.cn (F.W.); qjiang@cjlu.edu.cn (Q.J.); dingyanfei@cjlu.edu.cn (Y.D.); pzhch@cjlu.edu.cn (C.Z.); 2College of Engineering, Nanjing Agricultural University, Nanjing 210031, China; lihua@njau.edu.cn

**Keywords:** *Dendrobium officinale*, rapid in vitro propagation, low-temperature stress, gene expression, bioactive compounds

## Abstract

*Dendrobium officinale* is a valuable traditional Chinese herbal plant that is both medicinal and edible. However, the yield of wild *Dendrobium officinale* is limited. Adverse stress affects the growth, development, and yield of plants, among which low temperature is the primary limiting factor for introducing *Dendrobium officinale* to high-latitude areas and expanding the planting area. Therefore, this study aims to explore the variations in growth ability, cold resistance, and contents of bioactive compounds among different *Dendrobium officinale* strains. Four strains of *Dendrobium officinale* were selected as experimental materials and were subjected to low-temperature stress (4 °C). The agronomic traits, physiological indices, as well as the expressions of cold resistance-related genes (*HSP70*, *DcPP2C5*, *DoCDPK1*, and *DoCDPK6*) in the roots and leaves of *Dendrobium officinale*, were determined. The contents of bioactive compounds, including polysaccharides, flavonoids, and phenols were also measured. Compared with the other strains, Xianju had the highest seed germination and transplantation-related survival rates. Under low-temperature stress, Xianju exhibited the strongest cold resistance ability, as revealed by the changes in water contents, chlorophyll levels, electrical conductivities, enzyme activities, and expressions of the cold resistance-related genes. Additionally, the polysaccharide content of Xianju increased the most, while the stem flavonoid and leaf phenol contents were elevated in all four strains under cold treatment. Therefore, selecting excellent performing strains is expected to expand the planting area, improve the yield, and increase the economic benefits of *Dendrobium officinale* in high latitude areas with lower temperatures.

## 1. Introduction

*Dendrobium officinale* Kimura et Migo (*D. officinale*), known as a “life-saving herb”, is gaining popularity due to increased health awareness among people. However, the low reproduction and slow growth rates of *D. officinale* under natural conditions make wild resources unable to satisfy people’s demands. Artificial propagation has gradually advanced and is widely used to realize the sustainable use of *D. officinale* resources [1]. The current development trend is “*D. officinale* cultivation from the south to the north”. Consequently, cultivating *D. officinale* cold-resistant strains is particularly important and conducive to promoting the long-term development of the *D. officinale* industry. Advanced pharmacological and clinical trials have shown that *D. officinale* contains polysaccharides, flavonoids, alkaloids, dendrobium phenols, and other biologically active compounds, which show a variety of health-promoting effects, such as hypoglycemic [2], anti-tumor [3], immune regulatory [4], anti-oxidant [5], anti-inflammation [6], gastrointestinal protective [7], and others. Additionally, *D. officinale* is widely used in pharmaceuticals, dietary supplements, and the food industry with an increasing market demand [8]. The industrialization of *D. officinale* has developed rapidly in southern China and is distributed in Guangxi, Yunnan, Zhejiang, Fujian, Guizhou, Jiangxi, etc. These regions have formed a complete industrial chain from raw material cultivation to processing and production of health care products and have begun to have a specific industrial scale.

The germination rate of *D. officinale* seeds under natural conditions is low, which hinders large-scale sowing and planting. Seedling propagation technology is being rapidly adopted to meet the demands of both domestic and international markets [9]. This method can effectively shorten the nursery cycle and ensure the seedling quality, thereby accelerating the industrialization of *D. officinale*. Currently, *D. officinale* seedlings are generally propagated using cuttings and tissue culture. With the advanced use of the two seedling methods, the problem of seedling breeding has been solved, and low-carbon-utilizing seedling production has been realized. However, the traditional *D. officinale* cutting seedling nursery leads to poor seedling neatness, weak growth rate, and low one-time seedling yield, which is not conducive to the standardized scale cultivation of *D. officinale* [10]. To overcome these challenges, in vitro tissue culture has been used to improve plant survival rate [11], fundamentally accelerating the propagation speed of *D. officinale* [12]. In recent years, *D. officinale* tissue culture research has primarily included the selection of explants, the optimization of the culture medium, and other influencing factors.

To date, over 190 compounds have been isolated from *D. officinale*, primarily including polysaccharides, phenanthrenes, bibenzyls, phenols, amino acids, alkaloids, and others [13]. *D. officinale* polysaccharides can reduce malondialdehyde (MDA) content, elevate superoxide dismutase (SOD) levels, inhibit intracellular reactive oxygen species (ROS) production, and suppress H_2_O_2_-induced oxidative stress and apoptosis in H9c2 cells [14]. Numerous studies have confirmed the anti-tumor potential of *D. officinale*, with the primary underlying mechanism involving an enhancement in immune regulation and expression of tumor suppressor genes [15]. *D. officinale* polysaccharides (DOPs) could markedly induce interferon-γ (IFN-γ) and interleukin (IL)-10 secretion from the splenocytes of tumor-bearing mice by promoting splenocyte proliferation and enhancing the activity of natural killer (NK)cells and cytotoxic T lymphocytes (CTLs) [16]. This observation indicated that DOPs are potential anti-tumor drugs with immunomodulatory activities. Moreover, *D. officinale* stems have become major research material for Chinese and non-Chinese researchers. Recently, the *D. officinale* leaves have been shown to possess similar chemical components [17] and pharmacological activities as the stems [18]. Mice fed a high-fat diet showed a marked reduction in blood glucose levels and a significant improvement in glucose tolerance after treatment with *D. officinale* leaf aqueous extract (EDL) [19]. This observation indicated that the glucose-lowering effect of EDL may be related to the activation of the insulin signaling pathway. However, the pharmacological activity of *D. officinale* leaves has been less investigated in recent years. Therefore, investigating the efficacy of *D. officinale* leaves and developing novel products with the leaves as the main component is of great significance in realizing the development of new resources for *D. officinale.*

Low temperature is a major factor limiting plant growth and development, determining the geographical distribution of plants, and limiting the introduction of plant germplasm resources [20]. After sensing low-temperature-related signals, plants can improve their cold resistance by altering leaf cell structure, modulating physiological and biochemical responses, and modifying the expression of related genes [21]. Due to the excellent medicinal effects of *D. officinale*, it is well recognized by people, and the market demand is increasing annually. Currently, the “*D. officinale* cultivation from the south to the north” has become the development trend. Consequently, the cultivation of *D. officinale* with robust cold resistance is the key to improving the yield and expanding the planting area. *D. officinale* from different regions have varying cold resistance levels. In an environment with the same cold levels, the sprouting rate and stem thickness of the Zhejiang strains were the least affected and grew significantly better than other strains, the Jiangxi and Guangdong strains can recover after cold damage but show poor growth and the Yunnan strains were seriously affected by freezing, with a low rate of sprouting and weak growth [22]. Under low-temperature stress, the activities of SOD, catalase (CAT), and peroxidase (POD) enzymes of the antioxidant system of *D. officinale* were elevated to scavenge the reactive oxygen radicals, the chlorophyll content decreased while the electrolyte osmotic capacity increased, and MDA, soluble sugars, and proline were accumulated in large quantities to regulate the osmotic balance and improve the cold resistance ability. Only a few investigations on the molecular mechanism of low-temperature stress tolerance in *D. officinale* are available. The *DoWRKY5* gene of *D. officinale* was PCR-cloned, which clarified that it might play a vital regulatory role in the plant’s response to various abiotic stresses, such as low temperature [23]. *DcbHLH14* may regulate the transcriptional expression of the downstream functional genes and improve the stress resistance in *D. officinale* by relying on the abscisic acid(ABA)-signaling pathway to respond to low-temperature stress [24].

Despite its importance and excellent medicinal effects, the low reproduction and slow growth rates of *D. officinale* under natural low-temperature conditions make wild resources unable to satisfy market demands. The cultivation of *D. officinale* with robust cold resistance is the key to improving the yield and expanding the planting area. To solve this problem, the differences in growth ability among four different strains of *D. officinale* were investigated in this study. The *D. officinale* seedlings were subjected to low-temperature stress (4 °C) to study the cold resistance of different strains, and the differences in the content of bioactive components (polysaccharides, flavonoids, and phenols) with and without low-temperature stress were compared. This study identified strains of *D. officinale* with remarkably good growth ability, cold resistance, and quality. It was also found that the leaves of *D. officinale* had the same effective components as the stems, and the flavonoid content in the leaves was even higher. Additionally, the stem flavonoid and leaf phenol contents were elevated in all four strains under cold treatment. Therefore, selecting excellent performing strains is expected to expand the planting area, improve the yield, and increase the economic benefits of *D. officinale* in high latitude areas with lower temperatures. This study can provide a theoretical basis for more systematic and in-depth research and selection of cold-resistant strains in the future.

## 2. Materials and Methods

### 2.1. Plant Material and Culture Conditions

The *D. officinale* seeds used for the test were collected from planting bases in Taizhou, Zhejiang Province, China. The following strains were included: No. 1, Taizhou strain; No. 2, Xianju strain; No. 3, Fujian strain; and No. 4, Wenzhou strain (hereafter referred to as Taizhou, Xianju, Fujian, and Wenzhou, respectively). The experimental site, as well as the storage location of the sample, is at China Jiliang University, Hangzhou, China (longitude: 120.369036, latitude: 30.327401).

Tissue culture: The fruits of *D. officinale* were soaked in 75% ethanol for 30 s and rinsed with double distilled water (dd H_2_O) for a time. They were then soaked in 20% NaClO_3_ for 30 min and rinsed with dd H_2_O thrice. The sterilized fruits were placed in a sterile Petri dish. The head of the fruit was removed, revealing a small aperture at the top. The seeds were evenly sprinkled into the liquid medium in the culture flasks through the apertures and tightly sealed with a sealing film. The unopened seeds of the four strains were inoculated into the culture medium and placed in the tissue culture room. The medium consisted of Gamborg’s B-5 basal medium (B5), 0.5 mg/L 1-Naphthaleneacetic acid (NAA), 100 mL/L coconut water, 30 g/L sucrose, and 6.2 g/L agar [25]. After 2 months of cultivation, the subculture in the medium consisted of B5, 0.5 mg/L NAA, 0.2 mg/L Indole-3-butyric acid (IBA), 100 mL/L coconut water, 30 g/L sucrose, and 6.2 g/L agar [25]. The room temperature was maintained at 25 °C and a photoperiod of 12 h/day (h/d). The growth was regularly observed, photographed, and recorded during the test.

Determination of the physiological indices: The tissue-culture-raised seedlings of the four strains with consistent growth for six months were selected for low-temperature treatment. They were transferred to a plant growth incubator for continuous low-temperature stress treatment at 4 °C. After 0, 7, 14, 21, and 28 d of treatment, fresh leaves of 15 seedlings were cut, rinsed with dd H_2_O, dried on filter paper, flash-frozen in liquid nitrogen, and stored in an ultra-low-temperature refrigerator at −80 °C.

Collection method of plant material for testing the expression of cold-resistance-related genes: The leaves and roots of the seedlings incubated at 4 °C for 0, 12, 24, and 48 h were collected. They were rinsed clean with dd H_2_O, dried on filter paper, flash-frozen in liquid nitrogen, and stored in an ultra-low-temperature freezer at −80 °C.

Collection method of plant materials for determining the contents: The 10-month-old seedlings with consistent growth were exposed to low-temperature (4 °C) stress for 28 d. The seedlings grown at 25 °C were used as the control. The freshly obtained stems and leaves were wiped with 75% ethanol, dried at 60 °C, pulverized, and then passed through the standard No. 3 pharmaceutical sieve specified in the Chinese Pharmacopoeia, with a screen hole of 0.355 mm inner diameter and a mesh number of 50.

### 2.2. Determination of Agronomic Traits

After inoculation into the culture medium, the seeds of *D. officinale* swell and its embryo and seed coat enlarge continuously, changing from pale yellow to green, eventually forming young seedlings with true leaves. The seed germination rate was compared among four *D. officinale* strains at this stage based on the emergence of true leaves as a judgment basis.

Ten bottles were randomly selected for each strain, and 3 plants were randomly selected from each bottle. The plant height, stem thickness, root length, and plant weight were measured, and the number of roots was counted. The height was measured using a vernier caliper, with the base of the stem measured from the base to the tip, and the stem thickness measured for the third node starting from the top to the bottom. The root length was measured for the longest root of each plant.

### 2.3. Transplanting of the Tissue-Culture-Raised Seedlings

The 6-month-old tissue-culture-raised seedlings (*n* = 100), 4.0–5.0 cm in height, 2–3 mm in stem thickness, with ≥5 leaves and an average root length of 6.0–7.0 cm, were selected for hardening. They were transplanted to the natural environment of the laboratory and grown for 14 d. The seedlings were allowed to transition from a constant-temperature environment to an open one and then slowly adapted to the natural temperature and humidity. Before removing the seedlings from the bottle, the bottle was opened and placed in the laboratory. After 3–5 d, the seedlings were gently collected, the agar was washed off from their roots, and they were put in a ventilated and cooled place to dry. Once the roots turned white, the seedlings were then transplanted into the substrate, which was pine bark–substrate soil in a ratio of 1:1 (*w*/*w*). The seedlings were cultivated under natural conditions for 60 d, and the survival rate was calculated.

### 2.4. Determination of Water and Chlorophyll Contents

For this process, 0.15 g of leaves from the same parts of the seedlings of the four strains were weighed, dried at 105 °C for 3 h, transferred to a desiccator, and allowed to cool for 30 min. The material was reweighed precisely, dried at 105 °C for 1 h, allowed to cool, and weighed to a constant weight. The water content of the leaves was calculated based on the weight loss (%). Three replicates were employed for each sample.

By referring to the method of Knudson [26], 0.1 g of leaves from the same parts of the seedlings of the four strains were weighed, immersed in 7.5 mL 95% ethanol, and dark-treated for 24 h. The completely discolored leaves were removed by filtration, and the volume of the extract was made up to 10 mL with 95% ethanol. The blank was 95% ethanol, and the OD values of chlorophyll extract at wavelengths of 645 nm and 663 nm were determined using a UV-5100 spectrophotometer (Shanghai Metash Instrument Co., Ltd., Shanghai, China). Three replicates were obtained for each sample.

### 2.5. Determination of Electrical Conductivity

A total of 0.15 g of leaves from the same part of four strains of *D. officinale* were weighed, soaked in a 15 mL centrifuge tube with 8 mL of ddH_2_O for 8 h, and boiled in a boiling water bath for 10 min; the conductivity was then measured using a conductivity meter (DDS-307, Shanghai Yueping Scientific Instrument Manufacturing Co., Ltd., Shanghai, China) after cooling. Three replicates were employed for each sample.

### 2.6. Determination of MDA Content and CAT and POD Activities

MDA content and CAT and POD activities were determined using reagent kits from the Nanjing Jiancheng Biotechnology Research Institute as follows: microscale malondialdehyde (MDA) assay kit (A003-2), catalase (CAT) assay kit (A007-1), and peroxidase (POD) assay kit (A084-3). Three replicates were obtained for each sample.

### 2.7. Analysis of Expression Patterns of Cold Resistance-Related Genes

The total RNA from the roots and leaves of *D. officinale* was extracted using the TRIZOL method according to the instructions of the PrimeScript^TM^ II 1st Strand cDNA Synthesis Kit (TaKaRa). Reverse transcription was completed to obtain cDNA samples, which were stored in a −20 °C freezer for future use. The obtained cDNA sample was diluted three times for quantitative real-time PCR (qRT-PCR) experiments (Rotor-gene Q, Kaijie Enterprise Management (Shanghai) Co., Ltd., Shanghai, China). During the qRT-PCR operation, samples were added to a 72-well plate, and each sample was subjected to 3 biological replicates. The reaction system (10 μL) was as follows: TBGREEN II (TaKaRa) 5 μL, 0.4 upstream and 0.4 downstream primers each μL, cDNA 1 μL, and ddH_2_O 3.2 μL. Using the Rotor-Gene Q Series software 2.0.2, the reaction procedure was carried out as follows: 95 °C for 1 min, 95 °C for 20 s, 60 °C for 30 s, 72 °C for 20 s, and finally denature at 95 °C for 1 min, anneal to 60 °C, hold for 1 min, and then rise to 95 °C at a rate of 1 °C/s to draw a dissolution curve. Three replicates were obtained for each sample. The *β-actin* gene is used as an internal reference, and the primer sequence is shown in Table 1.

### 2.8. Determination of Polysaccharide Content

The tissue powder (from sieve No. 3) was precisely weighed to 0.3 g. The filtrate was placed in a 250 mL round bottom flask, 200 mL dd H_2_O was added, and the mixture was heat refluxed in a water bath for 2 h. After cooling, the aqueous solution was transferred into a 250 mL volumetric flask, and dd H_2_O was added to the scale of the isothermal liquid. The solution was shaken well and filtered. Precisely 2 mL of the filtrate was put in a 15 mL centrifuge tube, 10 mL of anhydrous ethanol was added, and the mixture was shaken well and refrigerated for 1 h. The filtrate was centrifuged at 4000 rpm for 20 min, and the supernatant was discarded. After precipitation, 8 mL of 80% ethanol was added, washed twice, and the supernatant was discarded. The precipitate was dissolved in hot water, cooled, isothermal liquid was added to a final volume of 25 mL, and then shaken well. The polysaccharide content was determined by the phenol–sulfate method [31]. The absorbance at 488 nm was determined using a UV-visible spectrophotometer (UV-5100, Shanghai Metash Instrument Co., Ltd., Shanghai, China). Three replicates were obtained for each sample.

### 2.9. Determination of Flavonoid Content

The tissue powder (from sieve No. 3) was precisely weighed to 0.3 g and extracted to determine the flavonoid content with reference to the method of Li [32]. The absorbance at 510 nm was determined using a UV-visible spectrophotometer (UV-5100, Shanghai Metash Instrument Co., Ltd., Shanghai, China). Three replicates were obtained for each sample.

### 2.10. Determination of Phenol Content

The tissue powder (from sieve No. 3) was weighed precisely to 0.3 g and added to 50 mL of 70% ethanol. The mixture was refluxed at 80 °C for 2 h to extract the phenol, then cooled and filtered, and made up to 50 mL with 70% ethanol, then shaken and mixed well. The phenol content was determined using the Folin–Ciocalteu colorimetric method [33]. The absorbance at 760 nm was determined using a UV-visible spectrophotometer (UV-5100, Shanghai Metash Instrument Co., Ltd., Shanghai, China). Three replicates were obtained for each sample.

### 2.11. Statistical Analysis

IBM SPSS Statistics 20 was used for data processing, and all data are represented as means ± standard deviations (SDs). The qRT-PCR data was analyzed using the 2^−∆∆Ct^ method [34].

## 3. Results

### 3.1. A Comparison of the Growth Abilities of the Four Strains of D. officinale

The seed germination rates of the four strains of *D. officinale* were recorded (Figure 1A), with Xianju having the best rate, followed by Wenzhou, and the lowest in Fujian, with only a small number of seeds germinating. Phenotypic differences among the four strains of *D. officinale* were observed at the seedling stage (Figure 1B), with the number of roots (Figure 1C) being markedly higher in Fujian than the others. The plant height in Wenzhou (Figure 1D) was appreciably more than that of the others, and the stem thickness was remarkably higher in Xianju than in Taizhou and Wenzhou (Figure 1E). The whole plant weight was markedly elevated in Xianju compared to Taizhou and Fujian (Figure 1F). However, there was no significant difference in the length of the roots among the four strains (Figure 1G). The transplantation survival rates of the four strains were ascertained, with Taizhou having the lowest (Figure 1H; Table 2 and Table 3). The plant growth in Taizhou was poor, with yellowish-green leaves, a shorter root system, and a visible withered and dead seedling phenomenon. Xianju had the highest survival rate and better plant growth, with robust stem segments and glossy green leaves.

### 3.2. Water and Chlorophyll Contents

Cold-induced damage dehydrates plant tissues, decreasing the leaf water content. Low-temperature stress caused a reduction in the free water content and improved the cold resistance in *D. officinale*. The water content of Taizhou reduced the most at 28 d of stress, reaching an average of 2.29%. However, Xianju showed the lowest change in water content, indicating superior cold resistance (Figure 2A).

Low-temperature stress damages chloroplast proteins, weakens photosynthesis, and decreases chlorophyll content [35,36]. After low-temperature stress, the chlorophyll contents of Taizhou and Fujian were initially elevated and then diminished. Compared with the untreated strains, the chlorophyll content of Taizhou reduced more markedly, reaching an average of 41.90%. In contrast, the chlorophyll content of Xianju was at first reduced and then increased, indicating that it possesses a certain degree of resistance to low-temperature stress and can maintain a higher chlorophyll content up to a certain period at low temperatures, which is conducive to its photosynthesis (Figure 2B).

### 3.3. Conductivity and MDA Content

The membrane system of plant cells is susceptible to disruption under low temperatures, leading to a remarkable elevation in the conductivity of crops, primarily due to excessive cytoplasmic exudation after the cell membrane was disrupted [37,38]. After 28 d of low-temperature stress, the conductivity of all four strains enhanced when compared with non-stressed controls. At 14 and 28 d of treatment, except in Taizhou, the conductivities of the other three strains decreased, indicating a certain degree of resistance and adaptability to adverse low temperatures. Among the four, Xianju showed the smallest increase, followed by Wenzhou, indicating better cold resistance (Figure 3A).

MDA is a lipid peroxidation product, and its content is used to assess the degree of lipid peroxidation. A low content indicates lower oxidative damage to the plant cell and organelle membranes and enhanced tolerance to low temperature [39]. The MDA content was diminished in all four strains, with Fujian showing the maximum decrease, followed by Xianju, indicating a certain degree of cold resistance (Figure 3B).

### 3.4. Determination of CAT and POD Activities

CAT and POD are antioxidant enzymes in plants, protecting them from oxidative damage. CAT and POD scavenge H_2_O_2_, impeding the accumulation of reactive oxygen radicals [40,41]. After low-temperature stress, CAT and POD activities were increased in Xianju and Fujian, were initially increased and then reduced in Taizhou, while they were gradually diminished in Wenzhou. Therefore, it can be inferred that Xianju and Fujian can minimize the degree of oxidation in *D. officinale* plants under low-temperature stress, can improve resistance, and possess a particular degree of cold resistance. CAT and POD activities were enhanced the most in Xianju, indicating a superior cold resistance (Figure 4).

### 3.5. Expression Analysis of Cold Resistance-Related Genes

The expression patterns of *HSP70*, *DcPP2C5*, *DoCDPK1*, and *DoCDPK6* after 48 h of low-temperature stress were analyzed using qRT-PCR, and they were elevated in comparison with the control group. The expression of *HSP70* increased by 1.16–2.68-fold, most apparently in Xianju (Figure 5A); the expression of *DcPP2C5* was enhanced by 1.10–11.35-fold, most obviously in Fujian, followed by 4.35-fold in Xianju, and by 2.79-fold at 24 h in Wenzhou. Subsequently, the expression declined but was still higher than the control group (Figure 5B). The expression of *DoCDPK1* peaked at 24 and 48 h, reaching a maximum of 20.03-fold in Xianju at 48 h and increased by 13.43–14.10-fold in the Taizhou and Fujian strains at 24 h. It then reduced but was still more than that in the control group (Figure 5C). *DoCDPK6* peaked, increasing by 4.27-, 3.95-, and 2.17-fold in the Xianju, Fujian, and Wenzhou strains at 48 h. It increased by 2.46-fold at 24 h in Taizhou, then declined, but was still higher than that of the control group (Figure 5D).

In the leaves of the *D. officinale* plants exposed to low-temperature stress, the expression of *HSP70*, *DcPP2C5*, *DoCDPK1*, and *DoCDPK6* increased at 48 h compared with the control group. The expression of *HSP70* reached a maximum at 12 and 48 h. The expression of *HSP70* continued to rise in Xianju and peaked at 12 h in the Taizhou, Fujian, and Wenzhou strains, and then declined, but was still higher than the control group (Figure 6A). The expression of *DcPP2C5* was at first enhanced, then fell and rose again in three strains, except in Wenzhou. It peaked at 13.42- and 5.85-fold in Fujian and Wenzhou at 12 h, and 17.37-fold in Xianju at 48 h (Figure 6B). The expression of *DoCDPK1* was higher in the leaves than in roots, indicating its tissue-specific expression, and was 11.92–56.21-fold higher than that of the control group at 48 h (Figure 6C). The expression of *DoCDPK6* also increased rapidly under low-temperature stress and peaked at 2.78–3.24-fold at 48 h (Figure 6D).

### 3.6. Polysaccharide Contents

Under normal growth conditions, compared with the control group, Xianju had the highest polysaccharide content of 22.17 ± 1.80%. The polysaccharide levels of Xianju, Fujian, and Wenzhou increased by 2.91%, 2.32%, and 0.53%, respectively, after low-temperature stress. The highest elevation in the leaf polysaccharide content was detected in Xianju; however, these contents in Taizhou decreased from 15.42 ± 1.12% to 13.01 ± 1.46% post-low-temperature stress (Figure 7A). Under normal growth conditions, compared with the control group, Xianju had the highest stem polysaccharide content at 24.38 ± 0.64%; those of Xianju and Fujian were elevated by 1.90% and 1.60%, respectively, and those of Taizhou and Wenzhou diminished by 1.64% and 2.70%, respectively, after the low-temperature stress treatment (Figure 7B).

### 3.7. Flavonoid Contents

Under normal growth conditions, Taizhou had the lowest leaf flavonoid content at 0.45 ± 0.02%, with the remaining three not differing much, at 0.80 ± 0.02%, 0.81 ± 0.01%, and 0.83 ± 0.03%. Compared to the control group, these contents did not change significantly in Xianju and Fujian after exposure to low-temperature stress; that of Taizhou was enhanced by 0.10%, and a decrease of 0.12% was seen in Wenzhou (Figure 8A). Under normal growth conditions, stem flavonoid contents were close to 0.22 ± 0.01% in Xianju and Fujian and 0.20 ± 0.01% in Taizhou and Wenzhou. Compared with the control group, these contents were elevated in the four strains post-low-temperature stress by 0.02%, 0.02%, 0.04%, and 0.03%, respectively (Figure 8B).

### 3.8. Dendrophenol Content

Under normal growth conditions, the phenol contents of Xianju, Fujian, Wenzhou, and Taizhou were 1.14 ± 0.01%, 1.15 ± 0.01%, 1.07 ± 0.01%, and 0.93 ± 0.01%, respectively. Compared to the control group, the phenol contents of the four strains were enhanced after low-temperature stress (Figure 9A). Under normal growth conditions, the contents of the Taizhou, Xianju, Fujian, and Wenzhou stems were 0.81 ± 0.01%, 0.75 ± 0.01%, 0.72 ± 0.01%, and 0.65 ± 0.02%, respectively (Figure 9B). Compared to the control group, after low-temperature stress, the stem phenol contents were reduced by 0.01% in Taizhou but enhanced by 0.02%, 0.03%, and 0.01% in the remaining three strains, respectively (Figure 9B).

## 4. Discussion

Under natural conditions of growth, *D. officinale* has a low reproduction rate and slow growth speed, with the wild resources unable to meet the people’s demands. Rapid artificial propagation is a gradually developing technology and is widely used to realize the sustainable use of *D. officinale* resources. A large number of high-quality *D. officinale* are cultivated using rapid in vitro propagation to satisfy the demand and resolve the market vacancy. From the perspective of seed germination, the seed germination rate was the best in Xianju, followed by Wenzhou, and the least in Fujian. The phenotypes of the seedlings of the four strains were observed in terms of plant height, stem thickness, number of roots, root length, and whole plant weight. The number of roots was markedly higher in Fujian, plant height in Wenzhou, stem thickness in Xianju, and whole plant weight in Xianju than the others, with no significant differences in the root length. In conclusion, Xianju and Wenzhou had better growth ability. The survival rates of the transplants of the four strains were statistically compared. After 60 d of transplanting, the survival rate was the lowest in Taizhou at 56%, with poor growth, yellowish-green leaf color, a shorter root system, and a noticeable withered and dead seedling appearance. The rate in Xianju was 77%, with good growth, a robust stem section, and leaves that were green and glossy. The rates of Fujian and Wenzhou were similar. The plant height and stem thickness in Taizhou were markedly lower than those of the other three; the leaf length of Xianju differed dramatically from those of the other three.

Low temperature is a major factor limiting plant growth and development, determining their geographical distribution, and restricting the introduction of germplasm resources. When sensing low-temperature-related signals, plants can improve their cold resistance by altering leaf cell structure, physiological and biochemical responses, and expression of related genes. Leaf cells are one of the most sensitive tissues to temperature stress. The leaves of plants susceptible to cold showed necrotic patches, yellowing and drying phenomenon, and even massive leaf drops under low-temperature stress, whereas the leaves of plants with a strong cold resistance maintained a bright green color, which indicated that the cold resistance performance varied considerably among the leaves of plants with differing cold resistance capabilities [42]. In this study, the leaves of 6-month-old tissue-culture-raised seedlings of four strains of *D. officinale* were used as experimental materials, and their physiological indices were measured after exposure to low-temperature stress. Low-temperature stress caused a reduction in the free water content and improved cold resistance, with Taizhou showing the most significant decrease at an average of 2.29%. However, the water content of Xianju had the most minor decrease, indicating its superior cold resistance ability. Moreover, plants with high chlorophyll content can adapt better to the low-temperature environment by suppressing the accumulation of toxic compounds and enhancing their antioxidant capacity. After low-temperature stress, the chlorophyll content of Taizhou was reduced the most, reaching an average of 41.90%. At 28 d of exposure, the chlorophyll content of Xianju was enhanced the most, suggesting a superior cold resistance ability by maintaining a higher chlorophyll content, which is conducive to photosynthesis.

When subjected to low temperatures, plants with superior cold resistance have more minor changes in their cell membranes and a lower rise in conductivity than plants with weaker resistance [43]. In this study, the conductivity of all four strains was elevated compared to that of the untreated controls. Xianju demonstrated the smallest increase, indicating superior cold resistance, followed by Wenzhou. Campos [44] suggested that the severity of damage to cell membranes, the level of lipid peroxidation, and the strength of the response of plants to adversity could be reflected by the MDA content. In this study, the MDA content of all four strains was diminished, most markedly in Fujian, followed by Xianju, which indicated a certain degree of cold resistance.

Under low-temperature stress, plants produce ROS that damage their membrane systems. The antioxidant defense system in plants can scavenge reactive oxygen radicals, thereby protecting normal physiological functions and enhancing their resistance to adversity. In this study, after low-temperature stress, CAT and POD activities increased in Xianju and Fujian, were initially enhanced but then reduced in Taizhou, and gradually diminished in Wenzhou. Accordingly, it can be inferred that Xianju and Fujian can suppress the oxidation degree under low-temperature stress, improve resistance, and demonstrate a particular cold resistance ability. The CAT and POD activities increased the most in Xianju, indicating its robust cold resistance. This observation was consistent with the findings of Shen [22], Tan [45], and Li [46].

The process of low-temperature signaling in plants involves the expression of the relevant cold resistance genes and the production of specific hormones, which can help mitigate the impact of cold damage. The HSP70 family of proteins is the most abundant and is vital in maintaining normal cellular life activities. Besides thermal stimulation, low temperature also induces the expression of heat stress proteins in large quantities [47]. In this study, the expression of the *HSP70* gene was up-regulated in the root and leaves of the tissue-culture-raised seedlings of *D. officinale* after low-temperature stress. Xianju showed the most evident rise, which was consistent with previous results [48], suggesting that *HSP70* may play a vital role in the cold resistance mechanism of *D. officinale*. PP2C is also involved in regulating low-temperature stress tolerance in plants [49]. In this study, the expression of *DcPP2C5* was up-regulated in response to low-temperature stress, which was also reported previously. Ca2+ acts as a second messenger in plant cells; low temperatures can induce a rapid increase in Ca2+ concentration in plant cell membranes, activating a complex signaling pathway and altering the expression of stress-related genes. CDPKs can function in the resistance of plants to stress by recognizing and participating in intracellular Ca^2+^-based signaling [50]. In this study, the expression of *DoCDPK1* and *DoCDPK6* was enhanced after low-temperature stress. *DoCDPK1* had the highest expression in leaves, suggesting tissue specificity, which was consistent with the findings of Sheng [29]. Based on the physiological indices under low-temperature stress and expression analysis of the related cold resistance genes, Xianju had the most superior tolerance, followed by Fujian, with Wenzhou and Taizhou having the weakest.

Advanced pharmacological and clinical trials showed that *D. officinale* contains polysaccharides, flavonoids, alkaloids, dendrophenols, and other biologically active compounds, with hypoglycemia, anti-tumor, immune regulatory, anti-oxidant, anti-inflammation, gastrointestinal protective, and other health-related effects [51,52]. Pharmacopeia stipulates that the stem is the medicinally important part of *D. officinale*. Consequently, during harvesting and processing, the non-medicinal parts, such as leaves, are usually discarded, wasting resources. Recent studies have shown that *D. officinale* leaves are rich in nutrients, more abundant in polysaccharides, total flavonoids, total phenolics, and alkaloids, and have a superior DPPH free radical scavenging capacity, with notably better antioxidant effect of the leaf alcoholic extracts [53]. Acute oral and genetic toxicity tests showed that the leaves as a food ingredient posed a low potential risk to human health and were safe for consumption [54].

Most current investigations have concluded that the polysaccharide levels of *D. officinale* affected its efficacy and quality [55]. In this study, the stems and leaves of tissue-culture-raised seedlings of four strains of *D. officinale* were used as materials to explore the differences in their chemical composition. The leaves and stems of Xianju had the highest polysaccharide content, and the quality of Xianju was the best, followed by Fujian. Flavonoid and phenol levels were markedly higher in the leaves than in the stems, as already illustrated [56]. Taizhou had the lowest flavonoid content, with few differences in the remaining three strains. The stem phenol content of Taizhou was slightly more than that of the other three strains, but the leaf phenol content was lower in Taizhou than in the other three.

Low temperature promotes the accumulation of polysaccharides in *D. officinale*, and the polysaccharide content of the cold-resistant strain was markedly higher than that of the non-cold-resistant strain. In this study, except in Taizhou, the polysaccharide contents of the other three strains were elevated post-low-temperature stress compared with the untreated, with Xianju demonstrating the most significant increase, followed by Fujian. This result indicated that Xianju had superior cold resistance, which was consistent with the findings of Shen [22]. The leaf flavonoid content of Taizhou increased, that of Wenzhou decreased, and that of Xianju and Fujian did not change remarkably. However, the stem flavonoid content was enhanced in all four, which was consistent with a previous study in tartary buckwheat [57]. The leaf phenol content in the four strains was elevated, and so was the stem phenol content in the three strains other than Taizhou. Based on multiple indicators, Xianju was determined to be most cold-resistant and superior in quality.

## 5. Conclusions

Studies on the response mechanisms of *D. officinale* to various abiotic stresses are minimal, and the mechanism of action underlying the response process to low-temperature stress is rarely reported. This study aims to explore the variations in the growth ability, cold resistance, and contents of bioactive compounds of different strains of *D. officinale*. The results show that the growth ability of the Xianju strain is the strongest, with the highest survival rate after transplantation as well as better plant growth. From the changes in water content, chlorophyll content, electrical conductivity, MDA content, and antioxidant enzyme activities, it can be seen that Xianju has the strongest cold resistance under low-temperature stress. qRT-PCR analysis demonstrated that the *HSP70*, *DcPP2C5*, *DoCDPK1*, and *DoCDPK6* genes could play a crucial role in the early cold stress response of *D. officinale* seedlings. Under low-temperature stress, the content of polysaccharides increased the most in Xianju, while the content of stem flavonoids and leaf phenolic compounds increased in all strains. These findings help to elucidate the mechanisms by which *D. officinale* responds to low-temperature stress and provide useful evidence for the selection of cold-resistant strains, the expansion of cultivation areas, and the improvement of quality and yield of the crop. 

## Figures and Tables

**Figure 1 foods-13-01467-f001:**
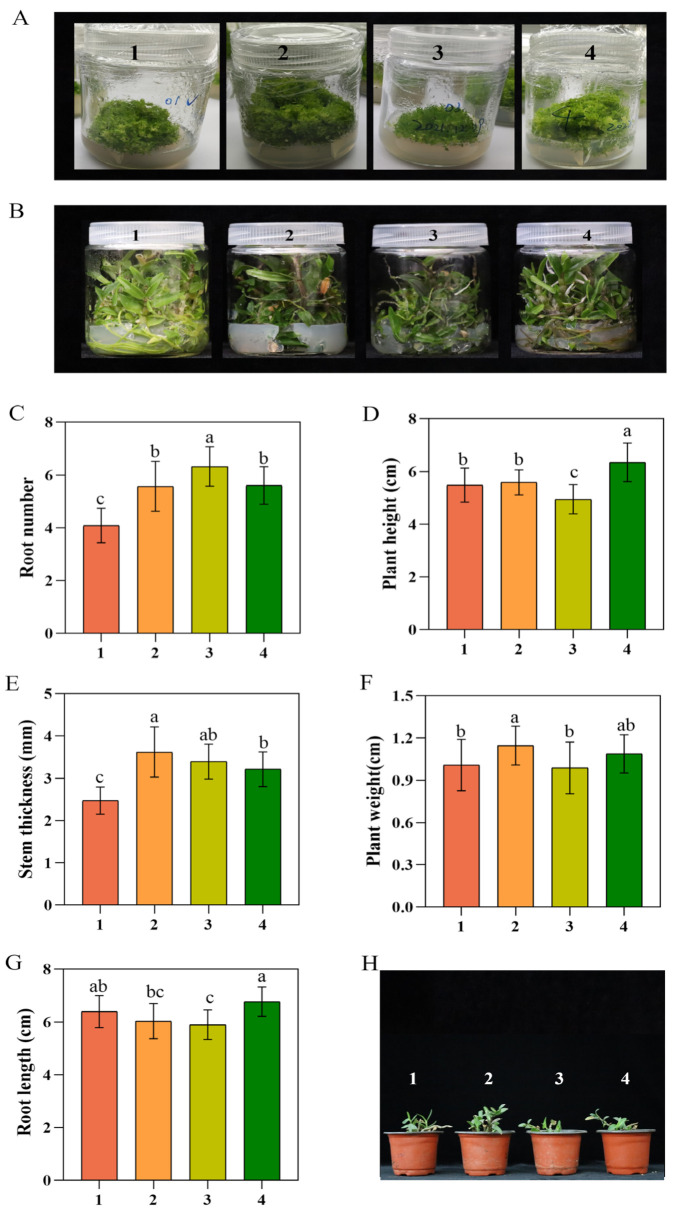
Seed germination rates (**A**), phenotypes of tissue-culture-raised seedlings (**B**), rooting number of 6-month-old plants ((**C**), *n* = 30), plant height of 6-month-old plants ((**D**), *n* = 30), stem thickness of 6-month-old plants ((**E**), *n* = 30), whole plant weight of 6-month-old plants ((**F**), *n* = 30), root length of 6-month-old plants ((**G**), *n* = 30), and the growth, 60 d after transplanting, (**H**) of the four strains of *D. officinale* (No. 1: Taizhou; No. 2: Xianju; No. 3: Fujian; No. 4: Wenzhou). The lowercase letters a, b, and c indicate statistically significant differences at *p* < 0.05, ascertained using the Duncan test.

**Figure 2 foods-13-01467-f002:**
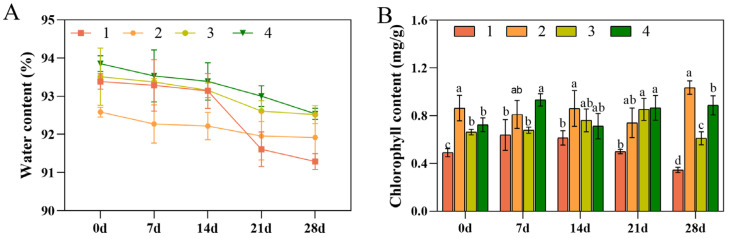
Changes in the water (**A**) and chlorophyll contents (**B**) of the four *D. officinale* strains at 0, 7, 14, 21, and 28 d after low-temperature stress (No. 1: Taizhou; No. 2: Xianju; No. 3: Fujian; No. 4: Wenzhou). Different letters indicate the statistically significant variations (*p* ≤ 0.05) among the germplasms. The error bars represent the standard deviations (SDs) of the means. Data from a mixed pool of leaves from each treatment was obtained (*n* = 3).

**Figure 3 foods-13-01467-f003:**
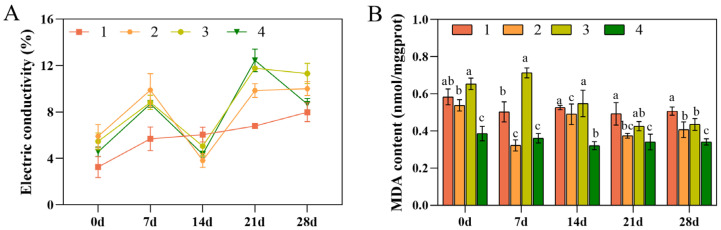
Changes in the conductivity (**A**) and MDA contents (**B**) of the four strains of *D. officinale* at 0, 7, 14, 21, and 28 d after low-temperature stress (No. 1: Taizhou; No. 2: Xianju; No. 3: Fujian; No. 4: Wenzhou). Different letters indicate statistically significant variations (*p* ≤ 0.05) among the germplasms. The error bars represent the standard deviations (SDs) of the means. Data from a mixed pool of leaves from each treatment was obtained (*n* = 3).

**Figure 4 foods-13-01467-f004:**
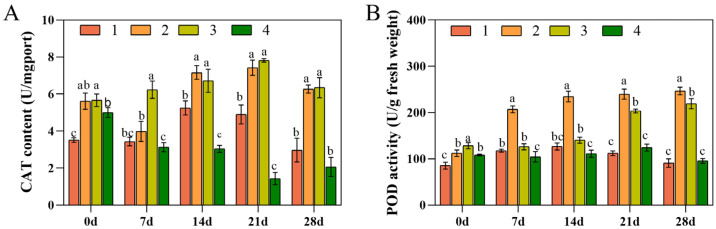
Changes in the CAT (**A**) and POD activities (**B**) of the four strains of *D. officinale* after 0, 7, 14, 21, and 28 d of low-temperature stress (No. 1: Taizhou; No. 2: Xianju; No. 3: Fujian; No. 4: Wenzhou). Different letters indicate the statistically significant variations (*p* ≤ 0.05) among the germplasms. The error bars represent the standard deviations (SDs) of the means. Data from a mixed pool of leaves from each treatment was obtained (*n* = 3).

**Figure 5 foods-13-01467-f005:**
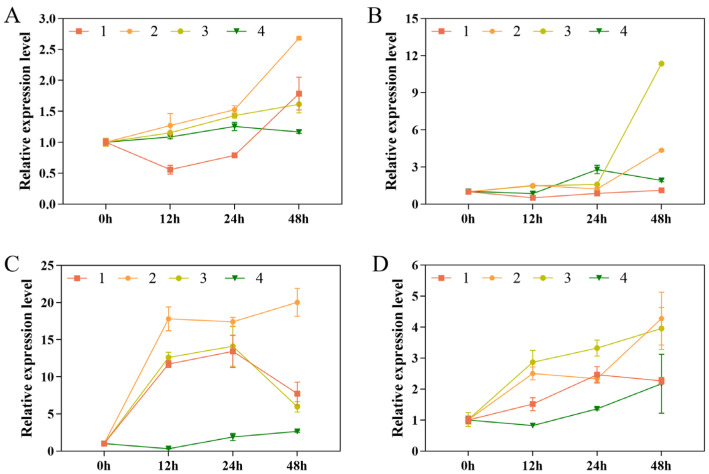
Relative expression of *HSP70* (**A**), *DcPP2C5* (**B**), *DoCDPK1* (**C**), and *DoCDPK6* (**D**) in the roots of *D. officinale* plants at 0, 12, 24, and 48 h of exposure to low-temperature stress (No. 1: Taizhou; No. 2: Xianju; No. 3: Fujian; No. 4: Wenzhou). The error bars represent the standard deviations (SDs) of the means (*n* = 3).

**Figure 6 foods-13-01467-f006:**
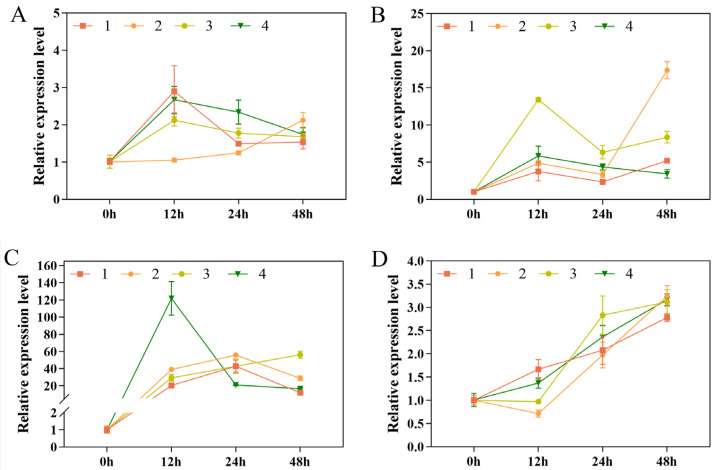
Relative expression of *HSP70* (**A**), *DcPP2C5* (**B**), *DoCDPK1* (**C**), and *DoCDPK6* (**D**) in the leaves of *D. officinale* plants after 0, 12, 24, and 48 h of exposure to low-temperature stress (No. 1: Taizhou; No. 2: Xianju; No. 3: Fujian; No. 4: Wenzhou). The error bars represent the standard deviations (SDs) of the means (*n* = 3).

**Figure 7 foods-13-01467-f007:**
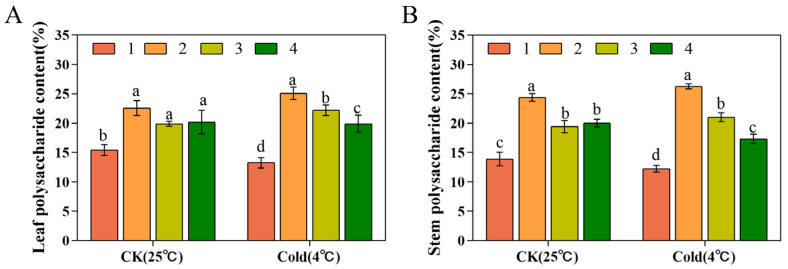
Variations in the leaf (**A**) and stem (**B**) polysaccharide contents of the four strains of *D. officinale* after low-temperature stress (No. 1: Taizhou; No. 2: Xianju; No. 3: Fujian; No. 4: Wenzhou). Different letters indicate statistically significant variations (*p* ≤ 0.05) among the germplasms. The error bars represent the standard deviations (SDs) of the means (*n* = 3).

**Figure 8 foods-13-01467-f008:**
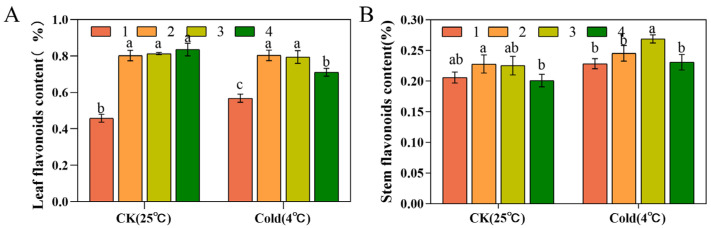
Differences in leaf (**A**) and stem (**B**) flavonoid contents of four *D. officinale* strains after low-temperature stress (No. 1: Taizhou; No. 2: Xianju; No. 3: Fujian; No. 4: Wenzhou). Different letters indicate statistically significant variations (*p* ≤ 0.05) among the germplasms. The error bars represent the standard deviations (SDs) of the means (*n* = 3).

**Figure 9 foods-13-01467-f009:**
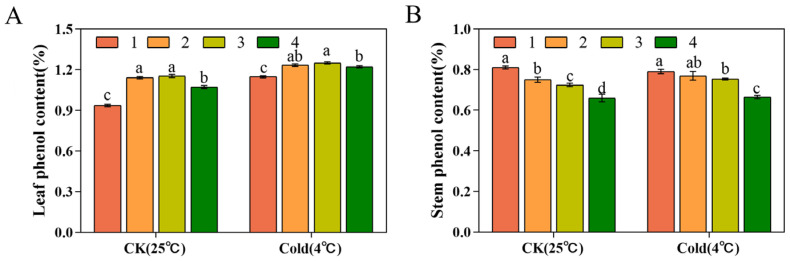
Differences in leaf (**A**) and stem (**B**) phenol contents of the four strains of *D. officinale* after low-temperature stress (No. 1: Taizhou; No. 2: Xianju; No. 3: Fujian; No. 4: Wenzhou). Different letters indicate statistically significant variations (*p* ≤ 0.05) among the germplasms. The error bars represent the standard deviations (SDs) of the means (*n* = 3).

**Table 1 foods-13-01467-t001:** qRT-PCR Primers.

Primer Name	Sequence (5′-3′)	References
*β-actin-F*	TTGTGTTGGATTCTGGTGATGGTGT	[27]
*β-actin-R*	TTTCCCGTTCTGCTGTTGTTGTGAA	
*HSP70-F*	TCTTACGACCTCTTTCTCAAGCCCT	[27]
*HSP70-R*	AATACCTATCGCTGGACCCTCTCCC	
*DcPP2C5-F*	CTTTGAAGATGTTGAGTTCCCAC	[28]
*DcPP2C5-R*	TGCTTTCCGCACTGAATCCT	
*DoCDPK1-F*	GAAAGATGCCGCTGTTGTAGTAAGA	[29]
*DoCDPK1-R*	CAATGTCGTGAAATCTCTTCTCTGG	
*DoCDPK6-F*	AAAGCGGTATGGTATTGAGGCAGAT	[30]
*DoCDPK6-R*	AAAATACCTTGTTCGGACTCTGCCC	

**Table 2 foods-13-01467-t002:** The survival rate of the *D. officinale* tissue-culture-raised seedlings of different strains.

Strain	1 Taizhou	2 Xianju	3 Fujian	4 Wenzhou
Survival rate (%)	56	77	68	70

**Table 3 foods-13-01467-t003:** Agronomic traits of the different strains of *D. officinale* tissue-culture-raised seedlings, 60 d after transplantation.

Strain	Plant Height (cm)	Stem Thickness (mm)	Leaf Length (cm)
1 Taizhou	5.03 ± 0.70 b	2.25 ± 0.24 b	3.31 ± 0.30 c
2 Xianju	6.21 ± 0.59 a	3.46 ± 0.31 a	4.55 ± 0.38 a
3 Fujian	5.28 ± 0.76 a,b	3.17 ± 0.38 a	3.32 ± 0.33 c
4 Wenzhou	6.11 ± 0.74 a	3.28 ± 0.40 a	3.75 ± 0.30 b

Different letters indicate statistically significant variations (*p* ≤ 0.05) among the germplasms. The error bars represent the standard deviations (SDs) of the means (*n* = 30).

## Data Availability

The original contributions presented in the study are included in the article; further enquiries can be directed to the corresponding author.

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
