# Peer review of "Variations in Cold Resistance and Contents of Bioactive Compounds among Dendrobium officinale Kimura et Migo Strains"

_foods, 2024, doi:10.3390/foods13101467_

Round 1

Reviewer 1 Report

Comments and Suggestions for Authors

Comments on the Quality of English Language

English language should be checked throughout the manuscript.

Reviewer 2 Report

Comments and Suggestions for Authors

A brief summary 

The manuscript reported contents of bioactive compounds of four different Dendrobium officinale Kimura & Migo and the evaluation of their cold tolerance.

General concept comments

The manuscript lacks of novelty, Dendrobium officinale is well-known and several manuscripts, including recent reviews have been reported. Manuscript’s results are reproducible. However, the introduction should be improved adding important information about aim of the work to the reader. The figures are correctly reported. The references are not correctly reported and too much self-references have been detected. The conclusions should be improved and more detailed. Data availability statements has been reported.

Specific comments 

1. The authors should underline the novelty of the study.

2. Dendrobium officinale Kimura & Migo has not been shortened and is repeated many times. 

3. Why was the term "strain" used?

4. Abstract is not clearly written; it should be changed to better clarify the introduction. Materials and methods and the results obtained. Xianju strongest ability of growth and cold resistance has been repeated many times.

5. The introduction should underline the aim of the work. 

6. Materials and methods should report the exact location of the sample.

7. The conclusions section should be improved and enriched with details based on the results obtained.

8. The references are not correctly reported, please change them following the authors guideline.

Comments on the Quality of English Language

Minor editing of English language required

Round 2

Reviewer 1 Report

Comments and Suggestions for Authors

The authors have modified the article according to the suggestions, and the manuscript has been significantly improved. Moderate English language corrections are still necessary, though.

Here are a couple of points that should be corrected to improve the flow of the text:

Line 123. Please erase the phrase” In general, the objectives and novelty of this study are as follows:”

Line 179. Please erase “a sieve No. 3 (refers to”

Lines 214-215. Please correct “The completely discolored leaves were removed by filtration, and the volume of the extract was made up to 10 mL with 95% ethanol.”

Comments on the Quality of English Language

Moderate English language corrections are still necessary.
